# Mode-Dependent Rectification for Stable PPO Training

## Abstract

Mode-dependent architectural components (layers that behave differently during training and evaluation, such as Batch Normalization or dropout) are commonly used in visual reinforcement learning but can destabilize on-policy optimization. We show that in Proximal Policy Optimization (PPO), discrepancies between training and evaluation behavior induced by Batch Normalization lead to policy mismatch, distributional drift, and reward collapse. We propose Mode-Dependent Rectification (MDR), a lightweight dual-phase training procedure that stabilizes PPO under mode-dependent layers without architectural changes. Experiments across procedurally generated games and real-world patch-localization tasks demonstrate that MDR consistently improves stability and performance, and extends naturally to other mode-dependent layers.

## 1. Introduction

Much of deep learning's progress has been driven by architectural innovations that enable large-scale optimization. Residual connections (He et al., 2016), transformer blocks (Vaswani et al., 2017), and normalization layers (Ioffe & Szegedy, 2015; Ba et al., 2016; Wu & He, 2018) have each played a central role in shaping modern neural networks. Among these, Batch Normalization (BatchNorm) (Ioffe & Szegedy, 2015) remains particularly influential, having improved optimization stability and training efficiency across a wide range of models. Despite the introduction of alternatives such as LayerNorm and GroupNorm (Ba et al., 2016; Wu & He, 2018), BatchNorm remains widely used in convolutional architectures.

In reinforcement learning (RL), normalization behaves differently. Unlike supervised learning, where data are drawn from a stationary distribution, RL data are generated by an evolving policy, inducing non-stationary state distributions. As a result, layers such as BatchNorm, whose behavior depends on data-driven statistics, are sensitive to discrepancies between training and evaluation distributions.

Prior work has reported mixed findings on the use of BatchNorm in RL. In off-policy settings, large replay buffers aggregate data from multiple past policies, introducing inter-policy distribution shifts even within a single batch (Bhatt et al., 2019; 2024). For PPO (Schulman et al., 2017), one might expect BatchNorm to behave more consistently, since training data are generated by the current policy. Yet empirical evidence remains mixed, with reports of both performance gains and degradation (Cobbe et al., 2019; Kanagawa & Kaneko, 2019).

In our experiments, incorporating BatchNorm into PPO resulted in catastrophic reward collapse. Stable training was only achieved by fixing BatchNorm layers in evaluation mode using pretrained (ImageNet) statistics. This observation motivates the central question of this work:

*Can instability induced by mode-dependent layers such as BatchNorm be systematically corrected to improve training stability and policy performance?*

To answer this question, we investigate the mechanisms underlying reward collapse in PPO when BatchNorm is used. Our analysis shows that instability arises from a mismatch between policies induced by training- and evaluation-mode behavior, amplified by distributional shift. We generalize this analysis to a broader class of mode-dependent layers and propose Mode-Dependent Rectification (MDR), a two-phase training procedure that introduces a deterministic rectification phase alongside standard updates to improve training stability. We evaluate MDR on PPO across procedurally generated games (Procgen) and two real-world patch-localization tasks. Our results show that MDR consistently stabilizes training and improves performance for both BatchNorm and dropout. Our main contributions are as follows:

- We identify and analyze how Batch Normalization induces reward collapse in PPO through policy mismatch and distributional shift.

- We extend this analysis to a broader class of mode-

[1]Anonymous Institution, Anonymous City, Anonymous Region, Anonymous Country. Correspondence to: Anonymous Author <anon.email@domain.com>.

Preliminary work. Under review by the International Conference on Machine Learning (ICML). Do not distribute.

dependent layers whose behavior differs between training and evaluation.

- We propose Mode-Dependent Rectification (MDR), a simple two-phase training procedure that stabilizes optimization without architectural changes.

- We empirically evaluate MDR across procedurally generated games and real-world visual tasks, demonstrating its effectiveness for BatchNorm and other mode-dependent layers such as dropout.

## 2. Background

### 2.1. Normalisation

Normalization layers are now standard components in deep neural networks, largely due to their ability to stabilize optimization and reduce sensitivity to parameter initialization. BatchNorm (Ioffe & Szegedy, 2015) introduced the use of minibatch statistics together with running estimates for inference, substantially accelerating training but making performance dependent on the assumption of i.i.d. batches and sufficiently large batch sizes. Batch Renormalization (Ioffe, 2017) alleviates this limitation by interpolating between batch statistics and running averages, improving robustness when batch distributions drift or when batch sizes are small.

A range of alternatives was subsequently proposed to remove this dependence on batch composition. Layer Normalization (Ba et al., 2016) computes per-sample statistics across all features, making it agnostic to batch size; it was originally introduced for recurrent networks and has since become standard in transformer architectures. Instance Normalization (Ulyanov et al., 2016), which normalizes each channel of each sample independently, was first used in image style-transfer models. Group Normalization (Wu & He, 2018) offers a middle ground by partitioning channels into groups and computing statistics within each group. Unlike BatchNorm, its performance is stable even when minibatches are very small, making it effective in detection and segmentation pipelines.

More recently, CrossNorm and SelfNorm (Tang et al., 2021) were introduced to improve generalization under distribution shift by explicitly manipulating channel-wise feature statistics. CrossNorm enlarges the effective training distribution by exchanging channel-wise means and variances between feature maps, thereby augmenting style variability during training. In contrast, SelfNorm learns to recalibrate these statistics through attention mechanisms with learnable weights. These methods employ distinct train–test behaviors as CrossNorm is applied only during training.

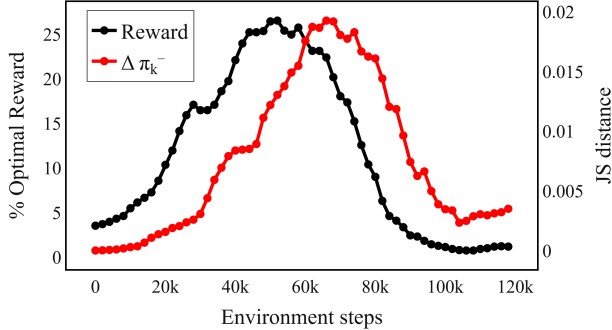

*Figure 1.* Reward collapse. Training curves when BatchNorm operates in training mode during optimization and evaluation mode during data collection. Rewards initially improve rapidly; as training progresses, the mismatch $\Delta\pi_k^-$ increases. Beyond a critical point, this growing mismatch coincides with a sudden collapse in performance.

### 2.2. Normalisation and Dropout in RL

The incorporation of normalization and mode-dependent layers in RL has received increasing attention in recent years. DroQ (Hiraoka et al., 2022) introduced the use of dropout layers together with layer normalization to regularize and "shrink" the Q-function ensemble used in REDQ (Chen et al., 2021) —the state-of-the-art at the time for sample efficiency through the usage of a large ensemble of Q functions. While DroQ maintains sample efficiency close to REDQ, it achieves substantially better computational efficiency, comparable to that of SAC (Haarnoja et al., 2018).

Another line of work studies the role of normalization in off-policy algorithms. (Bhatt et al., 2019) showed that naïvely applying BatchNorm is ineffective due to the action-distribution shift between the critic and the target networks (one updated under past policies, the other under the current policy). They proposed normalizing features on a mixture of on-policy and off-policy samples, combined with removing the target network, yielding significant improvements in TD3 (Fujimoto et al., 2018) and DDPG (Lillicrap et al., 2016). Building on this, they later introduced CrossQ (Bhatt et al., 2024), an off-policy method that uses wider critic networks and carefully applied BatchNorm and Batch Renormalization, and reported performance surpassing both DroQ and REDQ.

Beyond sample efficiency, (Li et al., 2024) investigated the use of CrossNorm and SelfNorm for improving generalization in RL, showing dramatic gains—for example, an increase from 14% to 97% success on CARLA. In contrast, most work in this area focuses on off-policy RL; results in on-policy settings remain mixed. (Cobbe et al., 2019) demonstrated that BatchNorm and dropout can improve generalization on the CoinRun benchmark (Cobbe et al.,

2019; 2020). However, (Kanagawa & Kaneko, 2019) found the opposite on their Rogue benchmark: BatchNorm underperformed relative to architectures that did not employ it, whereas LayerNorm performed better. Additional work (Voelcker et al., 2025; Lyle et al., 2024; Juliani & Ash, 2024) has further explored the role of LayerNorm in mitigating plasticity loss, reviving dormant neurons, and improving network stability.

To the best of our knowledge, no prior work has explained the discrepancy between these conflicting findings for Batch-Norm in on-policy RL. Our work provides the first explanation of the underlying mechanism through which Batch-Norm harms PPO in on-policy settings.

## 3. Method

### 3.1. Preliminary

**Batch Normalization.** BatchNorm (Ioffe & Szegedy, 2015) mitigates covariate shift in deep neural networks by normalizing activations within each mini-batch. Formally, given a mini-batch $B = \{x_1, x_2, \ldots, x_m\}$ of size $m$, the batch mean and variance are computed as:

$$\mu_B = \frac{1}{m} \sum_{i=1}^{m} x_i, \quad \sigma_B^2 = \frac{1}{m} \sum_{i=1}^{m} (x_i - \mu_B)^2 \quad (1)$$

The normalized activations are

$$\hat{x}_i = \frac{x_i - \mu_B}{\sqrt{\sigma_B^2 + \eta}}, \quad y_i = \gamma \hat{x}_i + \beta, \quad i = 1, \ldots, m \quad (2)$$

where $\eta$ is a small constant for numerical stability, $\gamma$ and $\beta$ are learnable parameters. At inference, when batch statistics are unavailable, BatchNorm uses global estimates of the mean and variance, $\mu_{\text{global}} = \mathbb{E}[D]$ and $\sigma_{\text{global}}^2 = \mathbb{V}[D]$ approximated during training via running averages with momentum $\mathcal{M}$:

$$\begin{aligned} \mu_{\text{r}} &= (1 - \mathcal{M}) \, \mu_{\text{r}} + \mathcal{M} \, \mu_B, \\ \sigma_{\text{r}}^2 &= (1 - \mathcal{M}) \, \sigma_{\text{r}}^2 + \mathcal{M} \, \sigma_B^2 \end{aligned} \quad (3)$$

**Proximal Policy Optimization.** Proximal Policy Optimization (PPO) (Schulman et al., 2017) optimizes a clipped surrogate objective:

$$L^{\text{CLIP}}(\theta) = \mathbb{E}_t \Big[ \min \Big( r_t(\theta) \hat{A}_t, \, \text{clip}(r_t(\theta), 1 - \epsilon, 1 + \epsilon) \hat{A}_t \Big) \Big]$$

$$\text{with} \quad r_t(\theta) = \frac{\pi_\theta(a_t \mid s_t)}{\pi_{\theta_{\text{old}}}(a_t \mid s_t)} \quad (4)$$

Here $\hat{A}_t$ is an advantage estimate and $\pi_\theta$ is a stochastic policy. The clipping term prevents large policy updates,

restricting changes to a trust region defined by $\epsilon$. The full PPO objective also includes a value-function loss and an entropy bonus:

$$L^{\text{PPO}}(\theta) = \mathbb{E}_t \Big[ L_t^{\text{CLIP}}(\theta) - c_1 \, L_t^{\text{VF}} + c_2 \, S[\pi_\theta](s_t) \Big] \quad (5)$$

$$L_t^{\text{VF}} = (V_w(s_t) - V_t^{\text{target}})^2$$

where $V_w$ is the critic, $S$ denotes the entropy bonus, and $c_1, c_2$ are weighting coefficients that balance the value and entropy terms.

### 3.2. BatchNorm Induced Instability

**Motivation.** During preliminary experiments with ResNet18, we found that using BatchNorm layers in the conventional supervised learning manner (by setting them to training mode during policy updates and evaluation mode during interaction) led to severe training instability. In some environments, the agent's reward collapsed rapidly. Stability was restored when BatchNorm was fixed to evaluation mode, using running statistics computed from ImageNet. This unexpected behavior motivated a systematic investigation into the interaction between BatchNorm and policy optimization.

**Formulation of the Phenomenon.** We define the following notation. Let $k$ index training steps, each consisting of a data collection phase followed by multiple parameter update iterations. Let $\pi_{\theta_k}$ denote the policy at step $k$, $\mu_k$ the corresponding state distribution, and $D_k \sim \mu_k$ the dataset collected at that step. Superscripts $-$ and $+$ indicate quantities before and after training within a step, respectively.

When BatchNorm operates in training mode, it uses mini-batch statistics $(\mu_B, \sigma_B)$, inducing the policy $\pi_{\theta_k}^{\mu_B, \sigma_B}$. In evaluation mode, BatchNorm relies on running statistics $(\mu_r, \sigma_r)$, yielding $\pi_{\theta_k}^{\mu_r, \sigma_r}$. During each step $k$, the dataset $D_k$ is collected under the evaluation-mode policy $\pi_{\theta_k}^{\mu_r, \sigma_r}$. We can then define the mismatch between the two Batch-Norm modes policies as:

$$\Delta \pi_k^- = \mathbb{E}_{s \sim \mu_k} \Big[ d\big( \pi_{\theta_k}^{\mu_B, \sigma_B}(\cdot \mid s), \pi_{\theta_k}^{\mu_r^-, \sigma_r^-}(\cdot \mid s) \big) \Big] \quad (6)$$

where $d(\cdot, \cdot)$ represents a probability distribution distance. After training, both the policy parameters and running statistics are updated, leading to

$$\Delta \pi_k^+ = \mathbb{E}_{s \sim \mu_k} \Big[ d\big( \pi_{\theta_{k+1}}^{\mu_B, \sigma_B}(\cdot \mid s), \pi_{\theta_{k+1}}^{\mu_r^+, \sigma_r^+}(\cdot \mid s) \big) \Big] \quad (7)$$

While the above expressions describe a single step, training unfolds over multiple steps, during which mismatches

compound recursively. In practice, $\Delta\pi_k^+$ propagates into $\Delta\pi_{k+1}^-$.

As training progresses, the policy $\pi_{\theta_k}$ evolves, inducing a corresponding shift in the state distribution. While early datasets satisfy $\mu_0 \approx \mu_1$, later distributions diverge, so that $\mu_k \not\approx \mu_{k+1}$. As a result, the running statistics (computed incrementally up to step $k$) no longer match the true batch statistics of the current step. Formally:

$$\mathbb{E}_{X\sim\mu_{k+1}}[X] \not\approx \frac{1}{k+1}\sum_{i=0}^{k}\mathbb{E}_{X\sim\mu_i}[X]\,, \qquad (8)$$

This amplifies the discrepancy $\Delta\pi_k^-$ at the beginning of each training cycle. Under severe shift, the model may exhibit training collapse, though the occurrence and severity depend on the task, environment dynamics, and the variability of the underlying state distribution.

When policy updates are aggressive, they can temporarily degrade performance. While PPO's clipped objective can partially correct such updates via negative advantages, collapse is typically accompanied by a substantial distribution shift between pre- and post-collapse rollouts (i.e., $\mu_{\text{pre-collapse}}$ and $\mu_{\text{post-collapse}}$). This shift amplifies instability, as well as the mismatch $\Delta\pi_k^-$, and hinders recovery.

We illustrate this behavior in Figure 1 by estimating $\Delta\pi_k^-$ as the average distance $d(\cdot, \cdot)$ computed over all mini-batches within each dataset $D_k$. For the distance metric, we employ the Jensen–Shannon (JS) divergence, which provides a symmetric and bounded alternative to the Kullback–Leibler (KL) divergence. As shown in the figure, the value of $\Delta\pi_k^-$ increases concurrently with the agent's reward, indicating that the statistical mismatch between BatchNorm modes grows as the policy improves. Eventually, this accumulation of mismatch leads to policy collapse, after which the divergence remains high due to the pronounced distributional shift between pre-collapse and post-collapse rollouts.

### 3.3. General Formulation

**Effect of Mode-Dependent Layers on the Policy Update.**

Let $N_\theta$ denote the policy network with parameters $\theta$, whose output defines the stochastic policy $\pi_\theta(\cdot \mid s)$. Suppose that certain layers within $N_\theta$ exhibit different behaviors between training and evaluation modes. This discrepancy induces a shift in the effective policy distribution at the level of individual states, which we define as:

$$\Delta\pi_\theta(s) = d\big(\pi_\theta^{\text{train}}(\cdot \mid s), \pi_\theta^{\text{eval}}(\cdot \mid s)\big) \qquad (9)$$

This state-wise formulation provides a state-level view of the step-level mismatch introduced in Eq. 6. During the interaction phase, rollouts are collected and the action proba-

bilities $\pi_\theta(a_t \mid s_t)$ are computed under the evaluation-mode policy $\pi_\theta^{\text{eval}}$, whereas during training, parameter updates are performed using the training-mode policy $\pi_\theta^{\text{train}}$. A direct consequence of $\Delta\pi_\theta$ is a shift between $\pi_\theta^{\text{train}}(a_t \mid s_t)$ and $\pi_\theta^{\text{eval}}(a_t \mid s_t)$ during optimization, which we denote by $\delta\pi_\theta(a_t \mid s_t)$.

Consequently, in the PPO surrogate objective, only the ratio term is affected as follows:

$$\begin{aligned} r_t'(\theta) &= \frac{\pi_\theta^{\text{train}}(a_t \mid s_t)}{\pi_{\theta_{\text{old}}}^{\text{eval}}(a_t \mid s_t)} \\ &= \frac{\pi_\theta^{\text{eval}}(a_t \mid s_t) + \delta\pi_\theta(a_t \mid s_t)}{\pi_{\theta_{\text{old}}}^{\text{eval}}(a_t \mid s_t)} \qquad (10) \\ &= r_t(\theta) + \delta r \end{aligned}$$

where $r_t'$ is the ratio actually used during optimization, $r_t$ is the ratio assuming mode-invariant behavior, and $\delta r$ captures the perturbation induced by mode-dependent layers.

We can interpret the consequences of this perturbation as follows:

**Trust-region shrinkage.** When the perturbation $\delta r$ tightens the clipping range, it effectively shrinks the trust region (Figure 2, clipped region). In this regime, the PPO objective still enforces conservative updates, and learning remains stable.

**Trust-region violation and instability.** In this case, the induced perturbation $\delta r$ results in policy updates that push $\pi_\theta$ outside the implicit trust region defined by $\epsilon$ in the clipped objective (Figure 2, unclipped region). We propose viewing this as a dynamic perturbation of the clipping boundary:

$$\epsilon' = \epsilon + \Delta\epsilon \qquad (11)$$

where $\Delta\epsilon > 0$ captures the deviation induced by $\delta r$. Under this interpretation, the instability introduced by different mode-dependent layers manifests as a perturbation of the clipping parameter $\epsilon$, as if the policy were optimized under a stochastic distortion of the trust region. This single quantity, $\Delta\epsilon$, captures the combined effect of these layers without modeling each one separately. When it becomes sufficiently large, the trust-region guarantee of PPO is violated, often resulting in unstable learning dynamics or catastrophic policy collapse.

### 3.4. Rectification

So far, we have shown that stochastic behavior induced by mode-dependent layers can be interpreted as a perturbation $\Delta\epsilon$ of the PPO clipping parameter, distorting the trust region

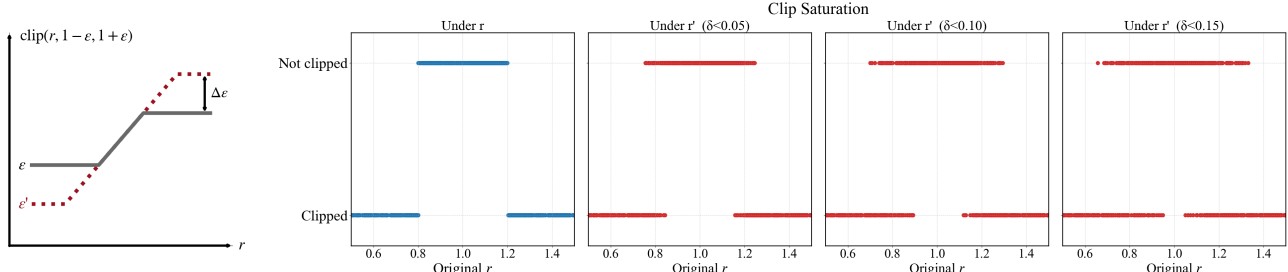

*Figure 2.* Effect of $\delta r$ and $\Delta\epsilon$ on PPO clipping. **Left**: Illustration of how a perturbation $\Delta\epsilon$ enlarges the effective clipping range of the PPO objective. **Right**: Clipping saturation as a function of the original ratio $r$. Blue points correspond to clipping under the unperturbed ratio $r$ ($\delta r = 0$), while red points show clipping under the perturbed ratio $r'$ with bounded noise $\delta r \in 0.05, 0.10, 0.15$. Increasing $\delta r$ progressively expands the effective clipping boundaries.

during optimization. Rather than attempting to model or suppress this perturbation at the level of individual layers, our goal is to correct the violations of the original trust region it induces, thereby preventing instability and policy collapse while preserving the benefits of mode-dependent layers.

MDR modifies the training procedure by splitting each optimization cycle into two phases:

1. **Standard update phase.** This corresponds to conventional training, performed for $\alpha_1 \times \frac{card(D_k)}{m}$ iterations, where $card(D_k)$ denotes the dataset size and $m$ the mini-batch size.

2. **Rectification phase.** An auxiliary training phase in which all layers are switched to their deterministic (evaluation) mode and optimized for $\alpha_2 \times \frac{card(D_k)}{m}$ iterations. Here, $\alpha_1$ and $\alpha_2$ are tunable hyperparameters.

The rectification phase acts as a corrective step that restores adherence to the original trust region after deviations introduced during the standard update phase. By optimizing the policy under deterministic layer behavior, it removes stochastic distortions of the clipping objective.

Notably, the entropy bonus in Equation (5) provides a principled corrective mechanism: when stochastic perturbations in the standard phase drive the policy beyond the original trust-region boundary, yielding an overly deterministic solution, the entropy term explicitly favors higher-entropy alternatives during the rectification phase.

This intuition can be formalized as follows. Consider a training phase in which the clipped PPO objective is evaluated with a perturbed bound $\epsilon + \Delta\epsilon$, producing an excessively deterministic policy $\pi$ that saturates the original clipping region. In this case, there may exist a more stochastic policy $\pi'$ such that:

---

**Algorithm 1** PPO with MDR

**Input:** policy $\pi_\theta$, value function $V_w$, mini-batch size $m$, phase coefficients $\alpha_1, \alpha_2$
Initialize policy parameters $\theta$ and value parameters $w$
**for** each training step $k = 1, 2, \ldots, K$ **do**
  **Interaction Phase:**
  Collect rollouts $\mathcal{D}_k$
  Compute advantages $\hat{A}_k$ and value targets $\hat{V}_k$
  **Training Phase:**
  Set network to `train` mode
  **for** $i = 1$ **to** $\alpha_1 \times \frac{|\mathcal{D}_k|}{m}$ **do**
    Sample mini-batch $B \sim \mathcal{D}_k$
    Update $\theta, w$ using PPO objective $L^{\text{PPO}}$
  **end for**
  **Rectification Phase:**
  Set network to `eval` mode
  **for** $j = 1$ **to** $\alpha_2 \times \frac{|\mathcal{D}_k|}{m}$ **do**
    Sample mini-batch $B \sim \mathcal{D}_k$
    Update $\theta, w$ using PPO objective $L^{\text{PPO}}$
  **end for**
**end for**

---

$$L^{\text{clip}}(\pi', \epsilon + \Delta\epsilon) < L^{\text{clip}}(\pi, \epsilon + \Delta\epsilon)$$
$$L^{\text{clip}}(\pi', \epsilon) = L^{\text{clip}}(\pi, \epsilon) \qquad (12)$$
$$\text{and} \quad \mathcal{S}(\pi') > \mathcal{S}(\pi)$$

where $\mathcal{S}(\pi)$ denotes the entropy of the policy. Thanks to the entropy regularization term, the PPO objective during the rectification phase favors $\pi'$ over $\pi$, effectively restoring stochasticity and stabilizing the optimization dynamics.

## 4. Experiments

### 4.1. BatchNorm with MDR

We first evaluate the effect of the proposed MDR procedure when applied to BatchNorm. Specifically, we compare three configurations during training: (i) BN, where BatchNorm

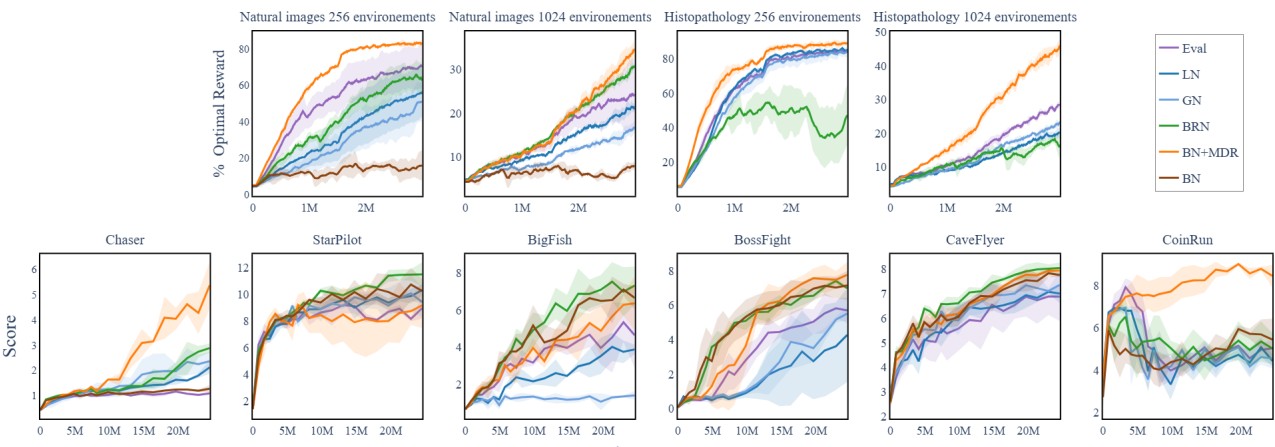

*Figure 3.* Performance comparison across patch-localization tasks (top) and Procgen games (bottom). Top: normalized reward expressed as a percentage of the optimal policy reward for natural-image and histopathology environments with 256 and 1024 environments. Bottom: average episode return (Score) on six Procgen games (500 environments). Shaded regions denote one standard deviation across three seeds.

layers operate in training mode using batch statistics; (ii) Eval, where BatchNorm layers are kept in evaluation mode throughout training; and (iii) BN+MDR, our proposed two-phase training procedure combining stochastic updates with a deterministic rectification phase.

To contextualize these results, we additionally consider architectures in which BatchNorm layers are replaced with alternative normalization schemes, including Layer Normalization (LN), Group Normalization (GN), and Batch Renormalization (BRN, used in training mode). These variants are implemented by directly substituting BatchNorm layers in the network backbone.

All experiments use a shallow ResNet-18 backbone obtained by removing the final block. Unless otherwise stated, models are initialized from ImageNet-pretrained weights, which we found to improve performance across all normalization variants consistently. Further details are provided in the appendix.

We benchmark all variants across six games from the Procgen benchmark (Cobbe et al., 2019; 2020): CoinRun, Chaser, StarPilot, BigFish, BossFight, and CaveFlyer, as well as two patch-localization environments:

1. A histopathology-based task introduced in (Mohamad et al., 2025).

2. A natural image variant constructed with high-resolution images from OpenImages (Kuznetsova et al., 2020).

The two image-based environments are shown to exhibit severe reward collapse when BatchNorm is trained in standard training mode (see Figure 1), making them suitable

testbeds for evaluating the proposed rectification procedure. The Procgen games provide a complementary setting with diverse visual statistics and controlled procedural variation, allowing us to assess whether MDR remains beneficial even when BatchNorm does not immediately fail.

We consider three experimental settings: Procgen games trained on 500 easy levels for 25M environment steps; and the two patch-localisation environments evaluated under 256- and 1024-environment settings, each trained for 3M environment steps. All experiments are repeated across three random seeds, each using a different set of environments.

For the Procgen benchmark only, we additionally evaluate a small set of implementation variants. Specifically, for BatchNorm-based methods (BN, BN+MDR, and BRN), we consider two momentum settings (0.1 and 0.01). For BN+MDR, we further evaluate two configurations with $\alpha_1 = 2\alpha_2$ and $\alpha_2 = 2\alpha_1$, denoted as MDR(2, 1) and MDR(1, 2), respectively. For clarity, we report the best-performing variant for each method in Figure 3.

Figure 3 (top) reports performance on the patch-localization tasks, expressed as a percentage of the optimal policy reward[1]. BN+MDR consistently achieves the highest performance across both natural-image and histopathology settings. BN fails on natural images and is omitted for histopathology due to an even more severe reward collapse.

Among alternative normalization schemes, BRN improves performance over BN in the natural-image setting, but performs poorly on histopathology. This contrast suggests that

---

[1]A small performance increase mid-training is due to a learning rate scheduler used in the patch-localization experiments.

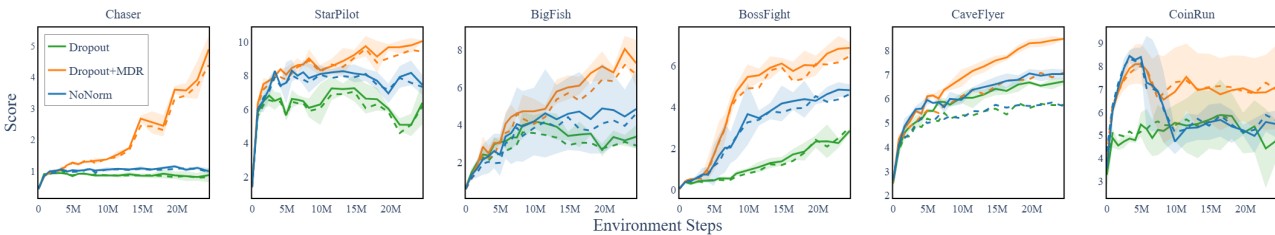

*Figure 4.* Dropout comparison on Procgen (500 easy levels). Solid lines denote training score, while dashed lines denote evaluation on held-out levels. Dropout is applied at 10%.

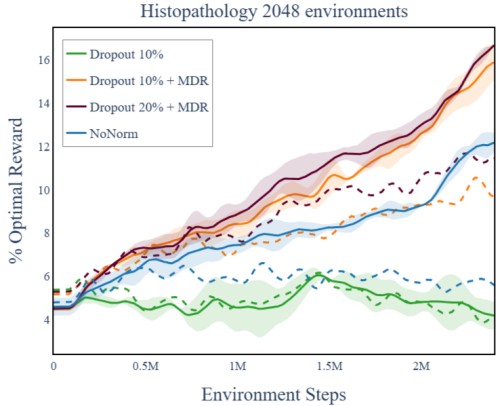

*Figure 5.* Dropout comparison on 2048 histopathology environments. In addition to the variants shown previously, we include an additional model with 20% dropout.

while renormalization can partially alleviate instability in settings with high visual variability, rectification provides a more reliable solution across domains.

Figure 3 (bottom) reports performance on the Procgen benchmark, measured as average episode return. BN+MDR achieves strong performance on Chaser and CoinRun, while underperforming on StarPilot; performance is otherwise comparable across the remaining games.

Across all Procgen environments, BRN consistently outperforms BN. We note that BRN and MDR target distinct sources of instability and are therefore complementary. While a full evaluation of their combination is beyond the scope of this work, preliminary results on histopathology indicate potential gains for BRN when trained under MDR.

### 4.2. Dropout with MDR

We evaluate whether the proposed rectification phase generalizes beyond BatchNorm to other mode-dependent layers, focusing on dropout. We compare three agents: (i) a baseline model with BatchNorm removed (NoNorm); (ii) a model where BatchNorm is replaced by dropout layers and trained using standard updates (Dropout); and (iii) the same dropout-based model trained with the rectification phase

(Dropout+MDR).

Experiments are conducted on the same 500-level Procgen setup used in the previous section. We report both training and test performance, with evaluation carried out on a separate set of 500 unseen levels. Results are averaged over three random seeds and shown in Figure 4.

This experiment evaluates: (1) how a policy trained with dropout compares to the baseline; (2) whether MDR improves the stability and performance of dropout-based policies; and (3) whether the rectification procedure interferes with the generalization benefits typically associated with dropout.

Figure 4 shows that agents trained with dropout alone (Dropout) consistently underperform the baseline (NoNorm) and exhibit reduced stability, with pronounced fluctuations and occasional performance degradation.

In contrast, incorporating MDR substantially improves performance when using dropout. The Dropout+MDR agent outperforms the baseline across all games, achieving higher and more stable returns. Importantly, improvements are observed in both training and test performance, indicating that the gains are not limited to the training environments.

We attribute this improvement to the regularization effect of dropout, which encourages generalization even across training levels. A related phenomenon has been reported in prior work (Cobbe et al., 2019; 2020), where increasing the number of training levels past a certain threshold leads to improved training performance. While the underlying mechanism differs, dropout in our case versus the number of environments in theirs, the observed behavior is consistent.

While Dropout+MDR improves test performance on Procgen, the resulting generalization gap is not clearly separable in this setting. To assess generalization behavior, we consider the histopathology patch-localization task, which exhibited strong overfitting.

We train agents on 2048 environments and evaluate them on 500 held-out environments. Figure 5 summarizes the results. As in the Procgen experiments, agents trained with Dropout+MDR achieve higher training returns than the base-

line.

Crucially, these gains are accompanied by improved test performance, resulting in a reduced generalization gap. In contrast, agents trained without dropout overfit to a large degree, achieving strong training performance but substantially degraded test returns. These results indicate that the generalization benefits of dropout are retained under rectification.

## 5. Conclusion

In this work, we showed that PPO training can be destabilized by mode-dependent layers such as BatchNorm, due to policy mismatch and distributional shift between training and evaluation modes. To address this, we introduced Mode-Dependent Rectification (MDR), a simple and general two-phase training procedure that restores trust-region consistency without modifying network architecture.

MDR improves both training stability and final performance across procedurally generated games and real-world patch-localization tasks. Our findings highlight MDR's robustness and its ability to generalize across different types of mode-dependent layers, including BatchNorm and dropout.

## Impact Statement

This paper presents work whose goal is to advance the field of Machine Learning. There are many potential societal consequences of our work, none which we feel must be specifically highlighted here.

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

## A. Reward Collapse

Figure 6 illustrates the relationship between reward and the policy mismatch measure $\Delta\pi^-$ for agents trained under different BatchNorm configurations (BN, Eval, and BN+MDR) and two PPO clipping values ($\epsilon \in 0.2, 0.05$). Each row corresponds to a training mode, and each column to a clipping value; results are shown for three random seeds.

To better isolate the collapse dynamics, these experiments are conducted with fewer transitions per dataset $D_k$, fewer optimization steps per step $k$, and only four histopathology environments. This setting amplifies distributional shifts and makes instability easier to observe.

Figure 1 shows a smoothed version of the top-right panel in Figure 6, using a running average to highlight the underlying trend.

**BN.** When BatchNorm operates in training mode during updates and evaluation mode during data collection, reward initially increases while $\Delta\pi^-$ grows steadily. At a critical point, reward collapses abruptly. After collapse, $\Delta\pi^-$ remains elevated, reflecting the persistent mismatch between pre-collapse and post-collapse state distributions ($\mu_{\text{pre}}$ and $\mu_{\text{post}}$), as discussed in Section 3.2. This behavior is consistent across both $\epsilon = 0.2$ and $\epsilon = 0.05$, indicating that tightening the clipping parameter alone does not prevent collapse.

**Eval.** With BatchNorm fixed in evaluation mode, training is substantially more stable. Occasional reward drops occur, particularly for larger $\epsilon$, but the agent reliably recovers. Correspondingly, $\Delta\pi^- = 0$ by definition.

**BN+MDR.** The BN+MDR configuration alternates between standard training-mode updates and the rectification phase. Here, temporary reward drops are followed by a spike in $\Delta\pi^-$. Crucially, these spikes decay rapidly, and the agent recovers without entering a collapse regime. This behavior is especially pronounced for $\epsilon = 0.2$, where BN fails but BN+MDR recovers. Lowering $\epsilon$ further reduces reward fluctuations.

## B. Rectification and Entropy

We analyze the effect of entropy regularization on the proposed rectification procedure using the histopathology patch-localization task. To accentuate instability, all agents are trained with fewer transitions per dataset $D_k$.

We compare BN+MDR and Eval configurations trained with and without an entropy bonus. Since disabling entropy can lead to sub-optimal behavior, we do not report mean performance across seeds. Instead, for each configuration, we select the two seeds exhibiting the largest reward fluctuations and aggregate them using a minimum operator, highlighting worst-case instability.

Figure 7 shows the resulting reward trajectories. Under Eval, entropy removal has little effect on stability. In contrast, under BN+MDR, removing the entropy bonus leads to substantially increased reward fluctuations, indicating that entropy regularization plays a stronger stabilizing role when correcting perturbations.

## C. Pipeline

### C.1. Environment

This section provides a brief overview of the environments used in our experiments, illustrated in the bottom part of Figure 8. We refer the reader to the original works for full environment specifications (Cobbe et al., 2019; 2020; Mohamad et al., 2025).

**Procgen.** Procgen is a benchmark of procedurally generated video games designed to evaluate generalization in RL (Cobbe et al., 2019; 2020). Each game supports "easy" and "hard" difficulty modes and defines environment-specific minimum and maximum achievable rewards. We evaluate six Procgen games:

- **CoinRun**: the agent navigates to collect a coin while avoiding obstacles and enemies ($R_{\text{min}} = 5$, $R_{\text{max}} = 10$).

- **StarPilot**: a side-scrolling shooter game ($R_{\text{min}} = 2.5$, $R_{\text{max}} = 64$).

- **CaveFlyer**: the agent navigates a cave network to reach a friendly ship ($R_{\text{min}} = 3.5$, $R_{\text{max}} = 12$).

- **Chaser**: a pursuit-based game inspired by *Ms. Pac-Man* ($R_{\text{min}} = 0.5$, $R_{\text{max}} = 13$).

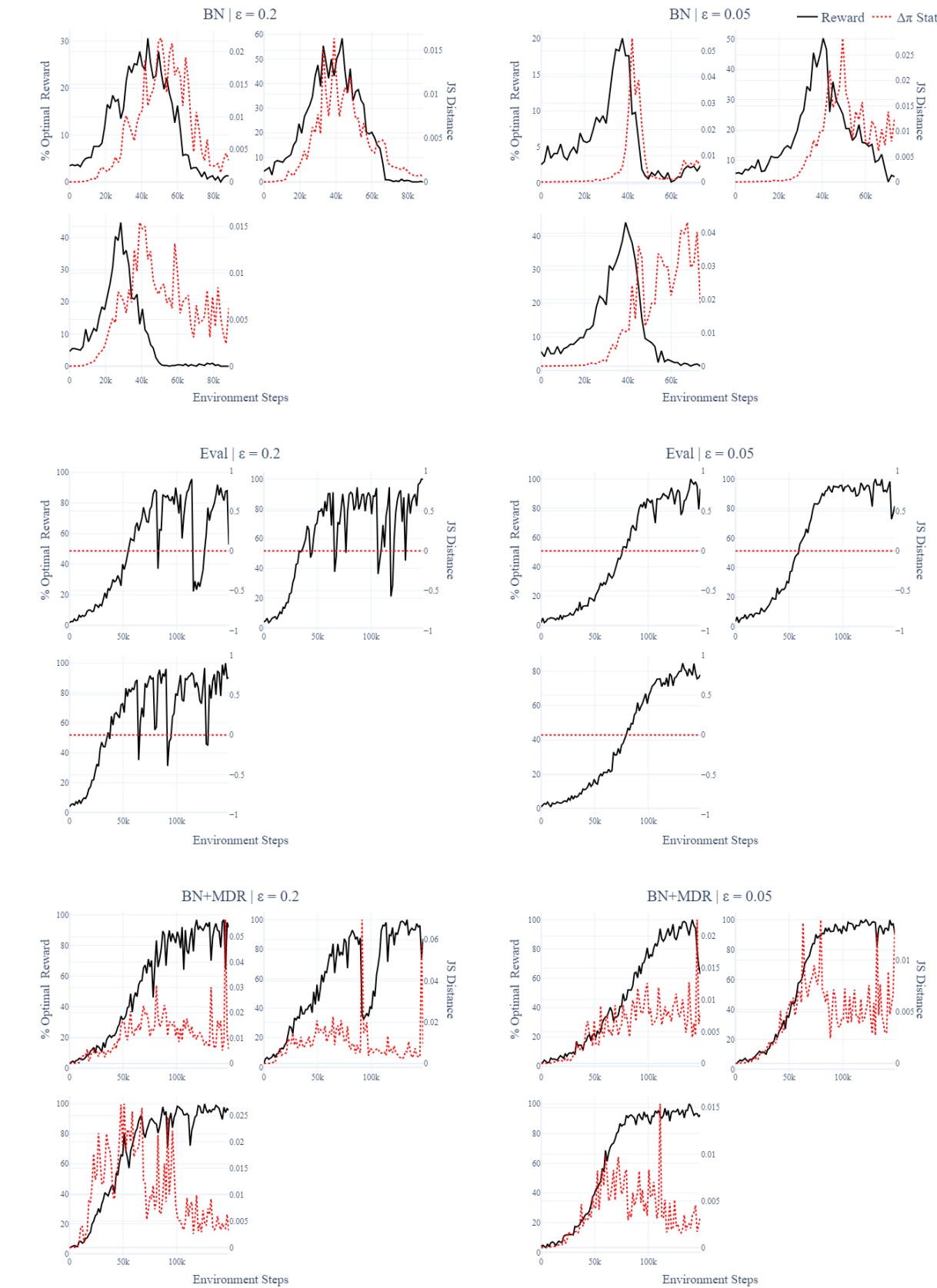

*Figure 6.* Reward and $\Delta\pi^-$ curves for agents trained in BN, Eval, and BN+MDR modes under two $\epsilon$ (clipping) values, over three seeds.

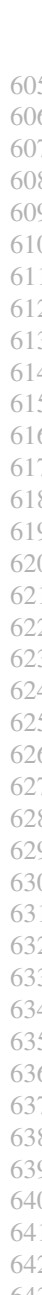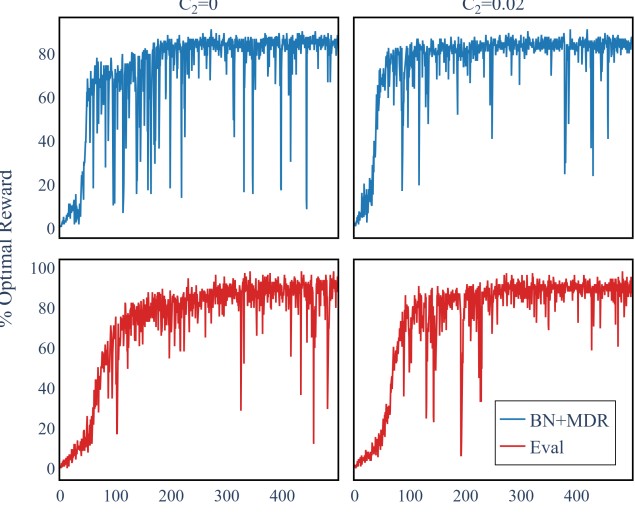

*Figure 7.* Effect of entropy regularization under rectification. Top row: BN+MDR; bottom row: Eval. Columns correspond to different entropy coefficients. Removing entropy leads to increased reward fluctuations under BN+MDR, while having a limited effect under Eval.

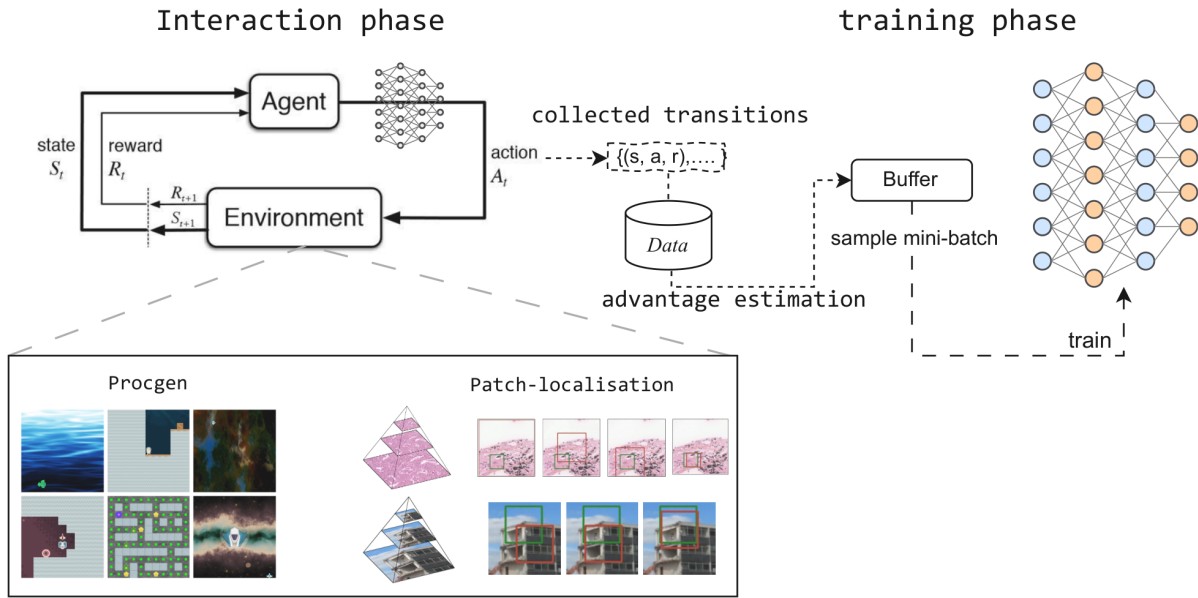

*Figure 8.* Pipeline (Sutton & Barto, 2018). During interaction, the agent collects experience by interacting with the environment. These experiences are then stored in a data-set. The advantage is estimated, and the agent is trained by sampling mini-Batches from the buffer. The bottom part shows the three visual environments used in this work.

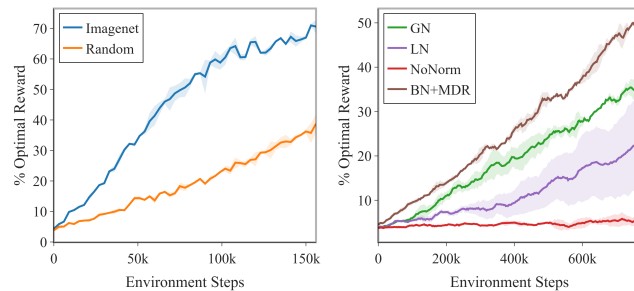

*Figure 9.* Effect of weight initialization on performance. Left: comparison of ImageNet and random initialization for BN+MDR. Right: comparison of normalization variants under random initialization.

- **BigFish**: the agent grows by consuming smaller fish while avoiding larger ones ($R_{\min} = 1$, $R_{\max} = 40$).

- **BossFight**: the agent must defeat a large enemy starship ($R_{\min} = 0.5$, $R_{\max} = 13$).

All Procgen environments share a $64 \times 64 \times 3$ RGB observation space and a discrete action space of 15 actions.

**Patch Localization.** Patch localization is a goal-conditioned visual navigation task (Mohamad et al., 2025) in which the agent must locate a target patch within a high-resolution image. Given an image of size $(m \times m)$, a target crop of size $(m/2 \times m/2)$ or $(m/4 \times m/4)$ is extracted and used as the goal. Both the full image and the target patch are resized to $112 \times 112$.

At each timestep, the agent observes a three-view input consisting of: (i) the target patch, (ii) a low-resolution view of the full image, and (iii) a local view centered at the agent's current position. The resulting observation has shape $3 \times 112 \times 112 \times 3$. The agent acts in a discrete action space of seven actions, corresponding to navigation and zoom operations that allow it to explore the image and localize the target.

### C.2. Architecture Details and Initialization

The architecture used throughout this work is derived from ResNet-18 (He et al., 2016), with the final residual block (Block 4) removed to reduce computational cost while preserving sufficient representational capacity. We refer to this variant as *shallow ResNet-18*. A single architecture is used across all environments.

We primarily adopt ImageNet-pretrained weights. Beyond improving performance across all methods, this initialization is essential for making BatchNorm in evaluation mode a meaningful baseline.

Figure 9 (left) illustrates the effect of ImageNet initialization on BN+MDR in the histopathology patch-localization task, showing a substantial improvement over random initialization. For completeness, Figure 9 (right) compares normalization variants under random initialization, where BN+MDR achieves the highest performance despite an overall reduction across all methods.

Table 1 summarizes the network architecture used for the actor and critic. Both networks share the same shallow ResNet backbone, followed by a two-layer MLP. For patch-localization tasks, three input views are processed independently using shared backbone weights and concatenated before the policy and value heads. For Procgen environments, the same backbone is used with a single input image, and the MLP dimensionality is adjusted accordingly.

## D. Hyperparameters

We use the Adam optimizer and Generalized Advantage Estimation (GAE) throughout all experiments (Kingma & Ba, 2017; Schulman et al., 2016). A summary of hyperparameters is provided in Table 2. Compared to the standard Procgen PPO configuration (Cobbe et al., 2020), we modify the entropy coefficient and disable reward normalization.

While the rectification phase is responsible for correcting policy violations, it is not merely a post-hoc adjustment step. It continues to optimize the PPO objective and does not increase the total number of training iterations per rollout. Instead, standard and rectification updates are interleaved by splitting the existing epochs between them, according to the ratio

*Table 1.* Shallow ResNet-18 architecture used in our experiments (Block 4 removed).

| Shared Backbone | |
|---|---|
| Input $(3, 112, 112, 3)$ | |
| Conv1 $(3, 56, 56, 64)$ | |
| Block 1 $(3, 28, 28, 64)$ | |
| Block 2 $(3, 14, 14, 128)$ | |
| Block 3 $(3, 7, 7, 256)$ | |
| Block 4 (removed) | |
| Average pooling $(3, 1, 1, 256)$ | |
| **Actor head** | **Value head** |
| Concatenate $(768)$ | Concatenate $(768)$ |
| FC $(768)$ | FC $(768)$ |
| FC $(768)$ | FC $(768)$ |
| Actions $(7)$ | Value $(1)$ |

$(\alpha_1, \alpha_2)$. As a result, MDR introduces no additional training overhead.

Although MDR introduces two new scheduling parameters $(\alpha_1, \alpha_2)$, we define them in proportion to each other (e.g., $\alpha_1 = 2 \cdot \alpha_2$ or $\alpha_2 = 2 \cdot \alpha_1$), effectively reducing tuning to a single degree of freedom. We found both $(1, 2)$ and $(2, 1)$ to work reliably across tasks.

In the patch-localization task, advantage estimates and value targets are recomputed every three epochs. For a total of nine epochs per rollout, this results in three full MDR rounds per rollout.

*Table 2.* Hyperparameters used for Procgen and patch-localization experiments. For MDR, epochs are split across standard and rectification phases as described in text.

| Hyperparameter | Procgen | Patch localization |
|---|---|---|
| Rollout size $|D_k|$ | 16384 | 3000 |
| Episode length | 1000 | 20 |
| Parallel environments | 64 | 4 |
| Epochs per update | 3 | 9 |
| Discount factor $\gamma$ | 0.999 | 0.99 |
| GAE parameter $\lambda$ | 0.95 | 0.95 |
| Value loss coefficient $c_1$ | 1.0 | 1.0 |
| Entropy coefficient $c_2$ | $1\times10^{-4}$ | $1\times10^{-4}$ |
| Batch size | 2048 | 128 |
| Learning rate | $5\times10^{-4}$ | $4\times10^{-5}$ |
| Weight decay | $1\times10^{-4}$ | $1\times10^{-4}$ |
| Gradient clipping | 1.0 | 1.0 |
| Advantage normalization | Yes | Yes |
| Input normalization (ImageNet mean/std) | No | No |
| Reward normalization | No | No |

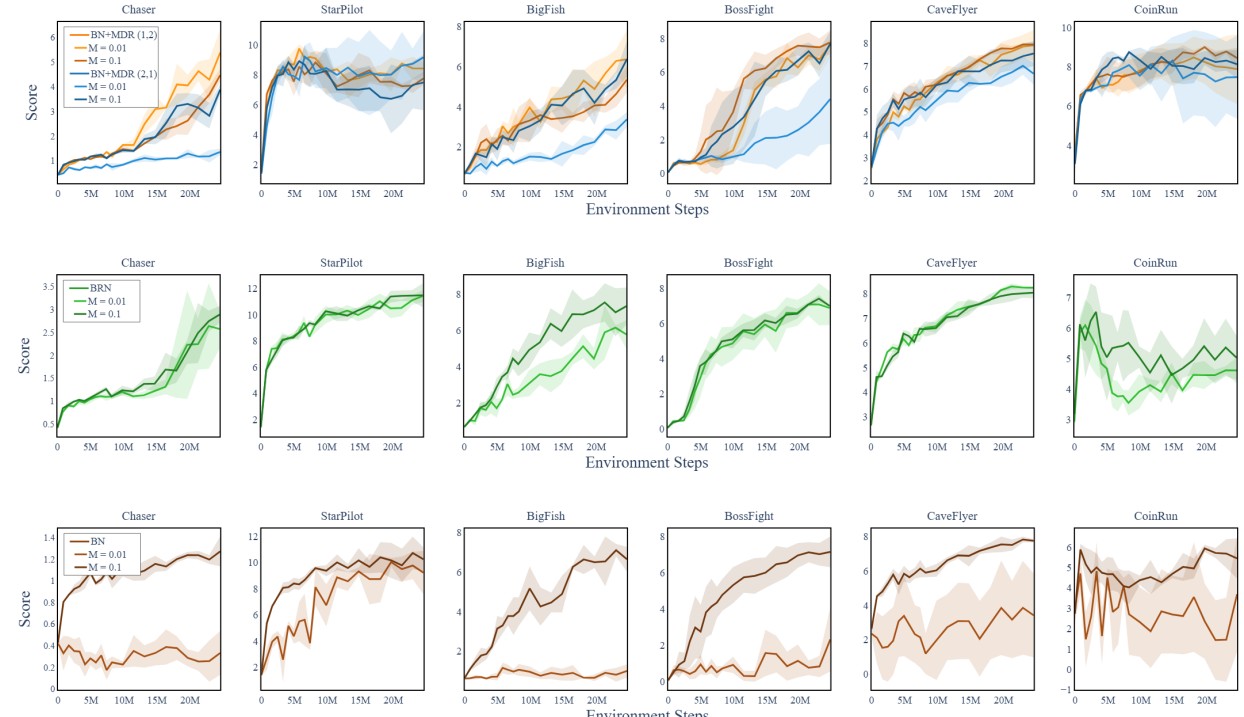

*Figure 10.* Additional visualizations for BatchNorm-based methods discussed in Section 4.1, including momentum variants for BN, BN+MDR, and BRN, as well as different standard-to-rectification phase ratios for BN+MDR.

