# OpenReview forum: "Mode-Dependent Rectification for Stable PPO Training"
_ICML.cc/2026/Conference — Submitted to ICML 2026_

### Official Review · Reviewer_LKrd · 2026-03-05

**Soundness:** 3
**Presentation:** 3
**Significance:** 1
**Originality:** 3
**Overall Recommendation:** 4
**Confidence:** 4

**Summary:**

The authors demonstrate that discrepancies between training and evaluation behavior in mode-dependent architectural components (such as Batch Normalization) lead to policy mismatch and reward collapse (in PPO training). Based on this observation, the authors propose MDR, a lightweight dual-phase training procedure that stabilizes PPO training. They claim that MDR might extend to other mode-dependent layers like Dropout.

**Compliance With Llm Reviewing Policy:**

Affirmed.

**Final Justification:**

I appreciate the additional experiments.

**Key Questions For Authors:**

I would raise the score if most questions mentioned in Strengths And Weaknesses are solved.

For now, the extent to which the method proposed in this paper is truly generalizable remains a question.

**Limitations:**

Yes.

**Strengths And Weaknesses:**

(1) Soundness: The paper is technically sound. The idea ( mode mismatch → trust region perturbation → reward collapse ) is well-reasoned. Figure 1 provides intuitive empirical support, and the ablations in Figure 6 further corroborate the analysis.

(2) Presentation: The paper is well-written and easy to follow, and the related work section covers the relevant literature adequately.

(3) Significance: The significance of this work is limited. First, the focus of this work is to stablize PPO training. While PPO is a widely used RL algorithm in the community, there are also many other RL algorithms and many other model architecture that use Batch Norm/Dropout. Can the proposed method be utilized on those as well? If not, the generalizabitlity of MDR can be questioned.

Second, it seems that only the results on BN have statistically significant improvements. The authors should add more experiment on Dropout, and they are encouraged to extend their methods on BRN and more mode-dependent architectural components.

 Last but not least, the authors did not test their methods on more benchmarks like DMC/Atari. This further undermines the significance of this work.

(4) Originality: I think its's very interesting and novel to model the difference in the behaviors of mode-dependent layers as $\Delta \epsilon$, and this can be a major strength of this paper. MDR itself looks quite straight forward.

---

> ### Author Rebuttal · Authors · 2026-03-31
>
> Thank you for your careful reading, thoughtful feedback, and for recognizing the technical soundness and novelty of the $\Delta \epsilon$. We would like to clarify the scope of our work and address questions regarding generalizability and significance.
>
> (1) Regarding generalization to other algorithms, the $\Delta \epsilon$ analysis can naturally extend to methods incorporating trust-region-like constraints (e.g., RePPO), where similar mechanisms are at play. While we do not experimentally validate these extensions, the underlying principle suggests that MDR could stabilize mode-dependent components in such settings. For other off-policy algorithms, applicability is less direct; however, prior work (e.g., CrossQ) shows that careful handling of BatchNorm can be beneficial, indicating that rectification strategies like MDR may still provide gains. Extending MDR beyond trust-region-based methods remains an open direction.
>
> From an architectural perspective, our analysis is not tied to a specific layer or network design (lines 207-210 and answer (1) to reviewer SRPZ). Empirically, we observed consistent improvements across two mode-dependent components, including BatchNorm and Dropout. Improvements to batchnorm and Renorm were more pronounced in patch localisation, where the layers exhibited collapse or large instability, suggesting that MDR captures a broader corrective principle rather than a layer-specific fix. We will clarify this scope more precisely in the revision.
>
> (2) We agree that the current paper does not establish broad generality. All experiments are conducted with PPO and a single architecture. While we show clear benefits for BatchNorm and promising improvements for Dropout, we do not exhaustively evaluate all layers or different architectures. To support the BRN claim (line 373), we provide an additional anonymized result here:
> https://anonymous.4open.science/r/repoanonym54355243/ (Fig. 1), comparing BRN with and without MDR. We will move this result to the appendix. Without MDR, PPO with Dropout fails to make meaningful progress compared to the baseline that excludes the layers (NoNorm, line 380), whereas with MDR, both training and test performance improve significantly. This demonstrates that MDR can address different mode-dependent instabilities. Dropout experiments, while more limited, confirm that MDR can stabilize other mode-dependent components, though train–test performance gap comparisons are only statistically significant in the patch-localization domain.
>
> (3) Regarding benchmarks, our selection aims to cover diverse environments (please refer to Reviewer 6Hon's response to question 3). While adding benchmarks such as Atari or DMC would further strengthen generality, our current suite already captures varied dynamics and clearly exhibits collapse behaviors.
>
> In summary, we recognize that MDR’s current experimental validation is limited to PPO (which, as noted by the reviewer, is still widely used) and a single architecture, and broader extension remains future work. We will revise the paper to clarify these limitations and explicitly position broader validation as an important direction.

---

> > ### Author Rebuttal · Reviewer_LKrd · 2026-04-01
> >
> > I thank the authors for their rebuttal. Overall, I am more positive with this work now.
> >
> > I maintain my current score primarily for two reasons:
> >
> > (1) The generalizability of the proposed method still remains open. I would like to point out that this is mentioned by all four reviewers, so it's a major weakness for this work. While some new experiments are added,  they are somewhat insufficient. For example, the authors mentioned ``In summary, we recognize that MDR’s current experimental validation is limited to PPO (which, as noted by the reviewer, is still widely used) and a single architecture, and broader extension remains future work. `` I appreciate that the authors have conducted experiments as through as they can  (``Our current experiments span over 100 runs and more than 200 GPU-days on an A40``), but they can add some smaller results on other benchmarks/algorithms/architectures to convince the reviewers. The authors are not expected to run all experiments exhaustively during the rebuttal (which may require too many computing resources), but they can report some positive intial results instead. If I receive the additional results, I am happy to raise my score to 4.
> >
> > (2) Despite this weakness, I believe this work still proposes a nice view about how to further improve mode-dependent architectural components. Therefore, I look forward to discussing the paper with the other reviewers and the AC first before finalizing my review.

---

> > > ### Author Response · Authors · 2026-04-08
> > >
> > > We appreciate the reviewer's continued engagement and positive reception of the paper. Following the suggestion to show additional results, we extended the empirical evidence during the rebuttal within the available time and computing budget.
> > >
> > > We prioritized extending across architectures and benchmarks, as these were the most feasible to incorporate quickly. The additional results are provided in the supplementary document:
> > > https://anonymous.4open.science/r/repoanonym54355243/extra_res.pdf.
> > >
> > > (1) Additional architectures: We evaluated MDR across multiple architectures on the patch-localization (histopathology) setting used in Figure 6 of the original submission. We selected this setting because:
> > > - It exhibits strong and consistent BatchNorm instability.
> > > - It is computationally more efficient, allowing broader coverage.
> > >
> > > We tested: RL baseline architectures: IMPALA CNN and Nature CNN, and deeper vision backbone: ResNet-50 (with and without ImageNet pretraining).
> > > Results (Figure 1 in the supplement) show that: All architectures except Nature CNN exhibit clear instability/collapse with BN, which is mitigated by MDR. The relative stability of Nature CNN is likely due to its shallower design.
> > > To further validate beyond the toy regime, we also ran IMPALA CNN on the 256 patch-localization Histopathology Environments (used in the main paper) for two seeds (Figure 3 in the supplement). These results still suggest gains when using MDR.
> > >
> > >
> > > (2) Additional benchmarks: We also extended experiments to Atari, using IMPALA CNN on (Breakout, Pong, Space Invaders) +Sticky actions (3 seeds, 5M steps).  In the default setting (Table 1, Figure 4), we observe that: Performance across baseline IMPALA, BN, and BN+MDR variants is very similar, making differentiation difficult.
> > > To stress the setting, we introduced a more aggressive hyperparameter configuration (higher learning rate and critic coefficient, Table 2): The baseline IMPALA degrades significantly, while BN and BN+MDR remain more stable.
> > > However, even in this regime, the difference between BN and BN+MDR remains moderate; we believe that longer training or a broader set of games might be required to fully expose the effect.

---

### Official Review · Reviewer_26AW · 2026-03-14

**Soundness:** 3
**Presentation:** 2
**Significance:** 2
**Originality:** 2
**Overall Recommendation:** 2
**Confidence:** 4

**Summary:**

This paper investigates why BatchNorm destabilizes PPO training in on-policy RL. The authors identify the root cause as a mismatch between train-mode and eval-mode BatchNorm behavior: data collection uses eval-mode statistics while parameter updates use train-mode statistics, inducing two effectively different policies. As training progresses and the state distribution shifts, this mismatch grows and eventually violates PPO's trust-region constraint, causing reward collapse. To address this, the authors propose Mode-Dependent Rectification (MDR), a two-phase training procedure that adds a rectification phase where all layers are switched to eval mode and the PPO objective is optimized again. MDR requires no architectural changes and adds no training overhead by splitting existing epochs between the two phases. Experiments on Procgen and two patch-localization tasks show that MDR stabilizes training and improves performance for both BatchNorm and dropout.

**Compliance With Llm Reviewing Policy:**

Affirmed.

**Final Justification:**

Most of my concerns have been addressed, but considering the overall quality of the paper, I can only raise my score to a 3.

**Key Questions For Authors:**

## Questions For Authors

**1.** LN performs comparably to BN+MDR in most experiments. Can the authors provide a concrete setting where BatchNorm must be retained and MDR offers a clear advantage over simply switching to LN?

**2.** Does MDR still prevent reward collapse under random initialization, or does it rely on pretrained running statistics to keep the initial mismatch small enough to recover from?

**3.** On Procgen games where BN does not collapse, is $\Delta\pi_k^-$ also smaller? Showing this would confirm that MDR's benefit scales with mismatch severity and clarify when the method is and is not expected to help.

**Limitations:**

No. The paper does not include a dedicated limitations section. The authors should discuss at least the following: (1) MDR's benefit appears conditional on BatchNorm causing severe collapse, and the conditions under which this occurs are not characterized; (2) the method's effectiveness under random initialization is substantially reduced, as shown in Figure 9, but this is not acknowledged as a limitation.

**Strengths And Weaknesses:**

## Strengths

**1.** The train/eval mismatch in BatchNorm is formalized as a perturbation $\Delta\epsilon$ on the PPO clipping parameter, providing the first mechanistic explanation that reconciles previously conflicting findings. The empirical evidence in Figure 1 and Figure 6 directly links $\Delta\pi_k^-$ growth to reward collapse.

**2.** MDR adds no training overhead by splitting existing epochs, and its two hyperparameters reduce to a single degree of freedom with low sensitivity to the exact values chosen.

---

## Weaknesses

**1.** The core motivation is weak. LayerNorm already performs comparably to BN+MDR in Figure 3, and is widely used in modern RL. The paper never establishes a concrete scenario where BatchNorm must be retained and LayerNorm is not viable.

**2.** Procgen results are inconsistent. BN+MDR clearly helps only on Chaser and CoinRun, and offers little benefit elsewhere. This suggests MDR is only useful when BatchNorm is already causing severe collapse, not as a general improvement.

**3.** The patch-localization task is a highly specialized setting where BatchNorm is known to be particularly problematic due to the large domain gap from ImageNet statistics. Using it as the primary demonstration risks overstating MDR's general applicability. Figure 9 further shows that without ImageNet pretraining, BN+MDR's advantage shrinks substantially, raising questions about how much MDR relies on pretrained running statistics rather than correcting the mismatch itself.

**4.** Only 3 seeds are used throughout. Given the high variance in training curves, many comparisons in Figure 3 fall within one standard deviation, making it hard to draw reliable conclusions.

---

> ### Author Rebuttal · Authors · 2026-03-31
>
> Thank you for the precise and very helpful feedback. We especially appreciate your focus on practical baselines and on pointing out when MDR is actually needed.
>
> (1) We would like to clarify the scope and motivation of our work. Our goal is not to argue that BatchNorm with MDR should replace LayerNorm, nor to establish a universally superior normalization strategy. Instead, we investigate whether addressing BatchNorm collapse is worthwhile, and whether a training-side correction such as MDR can make BatchNorm viable in settings where collapse occurs. Across our experiments, BN+MDR consistently outperforms BN when collapse occurs (Figure 3, top row, chaser, and coinrun). In most cases, BN or BN+MDR also outperforms LayerNorm (potentially influenced by the architecture choice). This supports our central claim: mitigating collapse is meaningful, even in contexts where LayerNorm is an alternative to mitigate collapse. Importantly, MDR is not specific to BatchNorm and can extend to other mode-dependent components such as Dropout (Figure 4).
>
> (2) Regarding Procgen, we agree that MDR provides gains when BatchNorm instability is present (e.g., Chaser, CoinRun). In environments where BN is stable, MDR does not degrade performance. Since MDR only splits existing epochs rather than adding training, it introduces minimal overhead (aside from one additional hyperparameter), making it a lightweight safeguard against collapse. We acknowledge that while we analyze the mechanism of collapse, we do not yet fully characterize when it will occur; we will explicitly list this as a limitation.
>
> (3) For the patch-localization setting, we intentionally consider a regime where BN is sensitive to domain shift. Figure 9 serves two purposes: (1) showing that pretrained initialization improves all methods, providing a meaningful baseline; and (2) demonstrating that BN+MDR still outperforms alternatives under random initialization, indicating that MDR directly mitigates mode mismatch rather than relying on pretrained statistics. We propose to substitute it with the clearer version found at the following anonymized link:
> https://anonymous.4open.science/r/repoanonym54355243/     (Figure 2).
> In this figure, dashed lines correspond to random initialization and solid lines to ImageNet initialization. In both regimes, BN+MDR consistently outperforms its counterparts.
>
> (4) Regarding the number of seeds (Our current experiments span over 100 runs and more than 200 GPU-days on an A40), we agree that our current empirical evidence is not sufficient to support strong generality claims. We will add a dedicated limitations section discussing: (i) the conditional nature of MDR’s benefits,  (ii) the limited number of seeds and benchmarks.

---

> > ### Author Rebuttal · Reviewer_26AW · 2026-04-04
> >
> > Most of my concerns have been addressed, but considering the overall quality of the paper, I can only raise my score to a 3.

---

> > > ### Author Response · Authors · 2026-04-08
> > >
> > > We thank the reviewer for the updated assessment and for addressing our rebuttal. We refer the reviewer to our response to Reviewer LKrd, which includes additional experiments conducted during the rebuttal period that may be relevant to the overall evaluation.

---

### Official Review · Reviewer_6Hon · 2026-03-14

**Soundness:** 2
**Presentation:** 3
**Significance:** 2
**Originality:** 3
**Overall Recommendation:** 4
**Confidence:** 3

**Summary:**

The authors present Mode Dependent Rectification, identifying issues with the standard deep learning modules that behave differently during training and inference, like batch norm and dropout, and mitigating them by proposing a dual-phase training procedure. The experiments are conducted on procedurally generated games and real world patch localization tasks.

**Compliance With Llm Reviewing Policy:**

Affirmed.

**Final Justification:**

increased by one pt given rebuttal, flagged one point that will still need some work

**Key Questions For Authors:**

I have covered my questions in the section above.

**Limitations:**

The paper includes a impact statement and mention no societal consequences worth highlighting here. It would be helpful if the authors can discuss some limitations associated with their current work, and future directions.

**Strengths And Weaknesses:**

1. Soundness
- The paper is well written, with a clear description of the existing problem, with relevant previous works cited, and a detailed empirical analysis along with an intuitive description that tries to explain why the training-inference mismatch leads to instability in PPO staining.
- A minor note I have regarding the paper's claims is that in Line 118 the authors claim that they "explain the discrepancy between these conflicting findings for batch norm", but their analysis is limited to empirically observed reward collapse. In my opinion, a complete explanation would include a theoretical argument for why the running statistics from training when used at inference time leads to this mismatch, explicitly quantifying the mismatch itself.
- Additionally, the two stage training does not perfectly fit the intuitive logic of increased clipping range, in the sense that during the first stage of training, we allow the clipping range to go further away, and only in stage 2 do we try to restore it back. This needs proper justification, why is the second stage training guaranteed to mitigate this issue, while not losing benefits of the original stage 1 training.
- In terms of evaluation, the intuition is well correlated, and the tasks do show improvements with the suggested change. My skeptcism here is based on the tasks themselves, and whether they are general enough to suggest that the proposed approach mitigates this issue completely. I would want the authors to clarify this further.
- The paper introduces two hyperparams $\alpha_1$ and $\alpha_2$, with the only ablation being choosing one as twice the value of the other and vice-versa. It is not clear where this choice comes from, and whether it would be agnostic to the setup - model, data, scale, etc.

2. Presentation
- The presentation of the paper is quite nice, with proper citations in place, and a clear description of their method. The narrative is easy to follow. My only suggestion here is to justify their approach better, grounding it to the intuition they explained earlier, as to why a corrective strategy makes sense here. It would also be helpful to place their method in a general context beyond their evaluation set, whether this corrective strategy to Batch Norm would be superior to other normalization variants like Layer Norm, which are standard practice nowadays.

3. Significance:
- The method seems to fix the issue with Batch Norm on the evaluated tasks, and also outperforms Layer Norm, Group Norm, etc. Similar is the case with dropout. If the model can be grounded better (see comments above) and validity of the method can be commented upon - keeping the model scale and type varying, and the task complexity changing, it would be significant to the larger RL community.

4. Originality:
- While two stage training is prevalent in deep learning to emphasize certain behaviours at certain stages of the training regime, the detailed analysis of the clipping range increase and an appropriately chosen method that mitigates the training-inference mismatch issue via this approach is not seen before. If general and stable enough, and the questions regarding the choice of the relative balance between two stages can be understood better, this could be adapted well in the RL community.

---

> ### Author Rebuttal · Authors · 2026-03-31
>
> we would like to thank the reviewer for the thoughtful and constructive review. We appreciate the point raised concerning hyperparamerers and motivation behind rectification phase, and the motivation behind environmnet choices for evaluation.
>
> (1) We agree that the current wording overstates the explanatory scope. Our goal is not to provide a complete theoretical explanation of the discrepancy, but rather a mechanistic and empirical account of how train/eval mismatch (BN) leads to instability in PPO. We will revise the text to make this scope more precise.
>
> (2) Regarding the motivation for the corrective strategy, we refer to our response to Reviewer (SRPZ) regarding the question of the motivation of the two-phase design. Concerning whether the second stage is guaranteed to mitigate the issue without losing the benefits of the first stage, our intuition is as follows: mode-dependent layers behave differently between training and evaluation, and the resulting increase in $\Delta \epsilon$ should be interpreted as a side effect of this mismatch, rather than the primary function the layers are designed to perform. This mismatch can lead to instability, and the role of the second stage is therefore not to undo the first phase, but to re-align the updated policy under evaluation-mode behavior, thereby reducing this discrepancy.
> We do not claim that rectification fully preserves all benefits of the first phase, and some loss may occur. However, our results suggest that the key improvements are retained. Experiments with both BatchNorm and Dropout combined with MDR show consistent improvements (Figures 3, 4, 5). In particular, Figure 5 (patch-localization task) demonstrates strong test-time performance (generalization) when Dropout is used with MDR, compared to the baseline excluding Dropout (NoNorm line 380-381).
>
> (3) Regarding the evaluation tasks, our goal was to cover diverse settings with different responses to BatchNorm, rather than focus on a single domain. We therefore included multiple domains: natural images, procedurally generated video games, and a medical imaging task (histopathology, where BN collapse is particularly pronounced). These tasks differ along several important axes: input structure (e.g., 64×64 RGB images for games vs. 112×112 multi-image inputs for patch localization), state-space complexity, horizon length, and reward dynamics. In particular, video game environments provide varied dynamics and reward structures across many levels, while patch localization introduces more complex state representations and domain shifts. We believe that this diversity allows us to evaluate the method across non-overlapping regimes, capturing a broad range of behaviors related to BN and Dropout instability. That said, we agree that extending the evaluation to additional benchmarks would further strengthen the generality claims, and we will clarify that the current selection is representative but not exhaustive.
>
> (4) Concerning the two hyperparameters, our design is motivated by splitting a fixed training budget between the two phases rather than adding overhead. While the second phase acts as a corrective step, it is also an optimization phase. The choice of 1:2 or 2:1 ratios reflects avoiding extreme imbalances where one phase becomes ineffective. We observed that such an interval worked well across tasks. However, we agree that a more exhaustive exploration of intermediate ratios and different setups would be valuable, and that the optimal split may depend on the model and task, which we acknowledge as a limitation.
>
> (5) Regarding the comparison with LayerNorm and broader positioning, we agree that this is an important question. However, determining when and where BN+MDR is superior to LayerNorm across architectures and tasks is beyond the scope of this work. Our goal is instead to assess whether addressing BN instability is valuable at all. Our experiments (on ResNet-style CNNs) show that once stabilized, BN can outperform LN in these settings. We will clarify this positioning and avoid suggesting broader claims.
>
> Finally, regarding all the comments. We agree that limitations should be made more explicit. We will add a dedicated section covering: (1) experiments restricted to a single architecture, with (2) limited hyperparameter ablation, and (3) limited benchmark scale.

---

> > ### Author Rebuttal · Reviewer_6Hon · 2026-04-04
> >
> > thank you for your detailed rebuttal. i believe a lot of my points are largely resolved and i will be happy to increase my score by one point. the most important detail i would want the authors to focus on is adding some more experiments on the training budget split, helping practitioners clearly see the trends of intermediate ratios.

---

> > > ### Author Response · Authors · 2026-04-08
> > >
> > > We thank the reviewer for the positive assessment and for raising their score. We agree that intermediate training budget ratios would be valuable for practitioners and will explore this in the final version. We also refer the reviewer to our response to Reviewer LKrd, which includes additional experiments conducted during the rebuttal period.

---

### Official Review · Reviewer_SRPZ · 2026-03-15

**Soundness:** 2
**Presentation:** 3
**Significance:** 1
**Originality:** 3
**Overall Recommendation:** 3
**Confidence:** 3

**Summary:**

This paper introduce an improvement of the existing PPO algorithm when applied to architecture with different training and eval behaviour by introducing an alternative training phases, to iteratively train in train mode and eval mode, which is called Mode-Dependent Rectification. The paper show experiments with ResNet-18 (to be more specific, with BN and dropout), on Procgen and two patch-localization environments. The paper also present a clipping range analysis to motivate this method.

**Compliance With Llm Reviewing Policy:**

Affirmed.

**Final Justification:**

Thanks for the authors' feedback. The rebuttal clears some of my concerns thus I raise the rating to 3. However, I still find the motivation of the paper is not very clear and the problem setting is not very practical, since usually modern networks do not involve modules whose behaviors differ between training and evaluation by design, while the differences introduced by nondeterminism implementation becomes more and more important in RL instead [*]; however, which is not easy to mitigate in MDR. In addition, the experiment seems too weak. Therefore, I recommend weak reject as the final rating.

[*] https://thinkingmachines.ai/blog/defeating-nondeterminism-in-llm-inference/

**Key Questions For Authors:**

1. How is the trust region analysis connects to the proposed method? And is there any deeper understanding on why we iterate on training and eval mode in the training will lead to better performance compared to the eval mode only(which introduces no difference actually)?

2. Is it possible to conduct eval replay like methods for the stochastic part of the network like dropout so we can compare this? Also, is there other kind of architecture that may be possible to benefit from the proposed methods except for BN and dropout? Especially those more used in current LLM/VLM model design would highly improve the practical applicability of the paper.

3. Another question is that what if we are not able to fully recover the eval mode behaviour in the training due to efficiency or infra incompatibility issue? How will this influence the method's effectiveness?

**Strengths And Weaknesses:**

### Strengths
- Model architecture behaves differently in training and eval time is a quite common cases; not limited to this paper's direct arch difference, often the time that training and inference infra difference may introduce difference in the training and inference time behavior, therefore this is quite an important problem.

- This paper present a simple method that can solve some kind of the training and eval difference cases, by simply iteratively training in training mode and inference mode. And the paper demonstrate the effectiveness of this simple method using ResNet18 on traditional RL tasks, and for the BN case, demonstrate stronger performance compared to the eval mode only training baseline.


### Weakness
- This method is not well motivated. I find it hard to draw connection between the motivation section that how the trust region analysis is related to the proposed methods.

- If we can replay the eval mode behavior in the training, this can serves as a strong baseline method to mitigate the training and eval mode behavior difference(e.g. router replay in LLM training), while the paper only compare this baseline method in the BN setting, while for the dropout setting, this is also an important baseline which is missing in the paper, weakening the whole arguments.

- Also, I think training and eval behavior difference can take many different shapes and I think not all of them can be solved by the proposed methods; I think a proper scope of the paper will be better to clearly state what kind of training and eval difference can be handled in proposed methods.

- In current transformer-based model designs, BN and dropout is less and less used, this further restrict current method's applicability to wider range or mainstream model architectures.

- The experimental settings seems too small and toy to validate the method's general applicability

---

> ### Author Rebuttal · Authors · 2026-03-31
>
> We thank the reviewer for the valuable comments. We address points (1) and (3) by starting directly from the reviewer’s observation that  ``training and evaluation behavior differences can take many different shapes.''  This is certain in mode-dependent layers, for example, BatchNorm, which uses different statistics in training and evaluation, and Dropout, which is stochastic during training.
>
> (1) To capture these effects in a general way, we introduced $\Delta \epsilon$  as a stochastic perturbation representing the difference in behavior between training and evaluation,  without assuming its shape, distribution, or evolution during training.
> This makes the formulation model- and layer-agnostic: as long as the train–eval difference manifests as a change in the output action distribution,  $\Delta \epsilon$ captures it.  Layer-specific train–eval shapes may influence the evolution or distribution of $\Delta \epsilon$, but by not making assumptions in this regard, the formulation remains general.
>
> The main motivation of MDR is not to explicitly model $\Delta \epsilon$ for each layer.  Instead, MDR aims to restore the trust-region constraint at the policy level. This is achieved by switching all mode-dependent layers to evaluation mode and re-optimizing the policy. This is done to re-anchors the policy $\pi_{\theta_{t+1}}^{\text{eval}}$   relative to the previous policy $\pi_{\theta_{t}}^{eval}$ used for the previous environment interaction. The rectification phase optimizes the network with $\Delta \epsilon=0$ and it directly addresses the trust-region violation incurred during training,  which is why it is important to place rectification immediately before the next experience collection step.
>
> We acknowledge that manually choosing $\alpha_1$ and $\alpha_2$ and keeping them fixed throughout training is a limitation. Different mode-dependent layers, or different steps of agent-environment interaction during training, may require varying amounts of rectification depending on the magnitude of $\Delta \epsilon$. A more adaptive correction that directly responds to the observed $\Delta \epsilon$ is a promising direction for future work. In the revision, we will make these points more explicit.
>
> (2–4) Concerning transformer-like architectures and router replay, we agree with the reviewer that BatchNorm and Dropout are less commonly used in mainstream architectures. However, smaller and more efficient CNN-like architectures are still widely employed in visual RL and other fields, such as medical research, where efficient feature extraction is important. We also recognize that layers such as Mixture-of-Experts (MoE)  can behave differently in training and evaluation modes, which makes them plausible candidates for our method. This suggests that our approach could be extended to architectures employing MoE. While our paper focused on visual RL with efficient backbones, layers like MoE present an important direction for future work, which we could explicitly highlight in the discussion.
>
> Regarding the router replay for Dropout, we would like the reviewer to clarify this point further. For BatchNorm in evaluation mode, the model is fixed in eval mode and uses imagenet running statistics, while only the scaling and shifting parameters remain trainable. Applying a similar approach to Dropout would effectively make the layer behave as an identity function. MDR is conceptually similar to router replay, but instead of evaluating both modes on the same instances, it performs a correction afterward using the deterministic evaluation behavior. Importantly, this correction is integrated into the optimization iterations themselves (see our response to Reviewer 6Hon, Question 4).
>
> (5) Concerning the experimental settings, we kindly refer the reviewer to our response to Reviewer 6Hon, Question 3.

---

> > ### Author Rebuttal · Reviewer_SRPZ · 2026-04-07
> >
> > Thanks for the authors' feedback. The rebuttal clears some of my concerns thus I raise the rating to 3. However, I still find the motivation of the paper is not very clear and the problem setting is not very practical, since usually modern networks do not involve modules whose behaviors differ between training and evaluation by design; instead, the differences introduced by nondeterminism implementation becomes more and more important in RL[*], however, which is not easy to mitigate in MDR. In addition, the experiment seems too weak. Therefore, I recommend weak reject as the final rating.
> >
> > [*] https://thinkingmachines.ai/blog/defeating-nondeterminism-in-llm-inference/
> >
> >
> > Clarify: in my comment, "router replay" in dropout means using the same dropout mask for the rollout and training steps within each RL step (instead of using "eval" mode for the rollout step), just like routing relay training for MoE [**].
> >
> > [**]  Zheng et al. Group Sequence Policy Optimization.

---

> > > ### Author Response · Authors · 2026-04-08
> > >
> > > We thank the reviewer for raising their score and for the clarification on router replay. We acknowledge the remaining concerns and will address the motivation and experimental scope more thoroughly in the final version. We also refer the reviewer to our response to Reviewer LKrd, which includes additional experiments conducted during the rebuttal period.

---

### Decision · Program_Chairs · 2026-04-30

**Decision:**

Reject

**Comment:**

### Summary of Contribution
The paper addresses instability in Proximal Policy Optimization training caused by Batch Normalization and Dropout, which exhibit different behaviors during training and evaluation. To mitigate policy mismatch and reward collapse, the authors propose Mode-Dependent Rectification, a dual-phase training procedure that introduces a rectification phase in evaluation to re-align the policy. Experimental validation was conducted  on procedurally generated games from the Procgen benchmark and real-world medical patch-localization tasks.
### Strong Points
Reviewers highlighted the significance and importance of addressing behavior discrepancies between training and evaluation, noting that such differences are a common issue in various model architectures and infrastructure (Reviewer SRPZ). The paper is also described as well-written and easy to follow, providing a clear description of the problem and the proposed method (Reviewers 6Hon, 26AW, and LKrd). Multiple reviewers appreciated the novel mechanistic explanation and theoretical formalization of the train/eval mismatch as a perturbation on PPO clipping. (Reviewers 26AW and LKrd).  The proposed Mode-Dependent Rectification (MDR) was noted as being simple, lightweight, and straightforward (Reviewers SRPZ, 26AW, and LKrd).
### Weak Points
Reviewers identified several weaknesses in the paper related to limited applicability, narrow experimental scope, and weak motivations:

- **Limited Practicality and Applicability:** Reviewers pointed out that modern architectures and current LLMs/VLMs rarely use BatchNorm or Dropout, which restrictis relevance (Reviewers SRPZ and 26AW). The additionally note that LayerNorm is, in fact, already a standard and effective alternative in RL that performs comparably to the proposed method without requiring specialized training phases (Reviewers 6Hon and 26AW).
- **Narrow Experimental Validation:** The initial evaluation was criticized for being conducted on a small number of benchmarks and "toy" settings, missing standard environments like Atari or DeepMind Control Suite (Reviewers SRPZ, 6Hon, and LKrd). Furthermore, the reliance on only three seeds and specialized tasks like medical patch-localization raised concerns about high variance and overstated generality (Reviewers 26AW and 6Hon). Additional experiments on Atari environmets were added in rebuttal, which was appreciated by Reviewer LKrd. However these new, preliminary results do not convincingly demonstrate stability of BN+MDR over baselines alone.
- **Underdeveloped Motivation and Theory:** The connection between the trust-region analysis and the proposed two-phase rectification procedure was unclear (Reviewers SRPZ and 6Hon). Reviewers also noted the lack of a formal theoretical quantification of the train/eval mismatch and a failure to characterize specific conditions under which BatchNorm is preferable to other normalization techniques (Reviewers 6Hon and 26AW).
- **Insufficient Hyperparameter Ablation:** The paper did not provide enough justification or exploration for the **fixed ratios** of its two hyperparameters, leaving it unclear how sensitive the method is to these choices across different scales and models (Reviewers 6Hon and 26AW).
- **Inconsistent Results:** Gains were noted to be inconsistent across tasks, with the method appearing primarily useful only in specific scenarios where severe reward collapse was already occurring (Reviewer 26AW). Statistical significance was also questioned for Dropout (LKrd).
### Recommendation
Despite the positive aspects of this submission identified by the reviewers, there was little enthusiastic support on their part for accepting the paper. Questions related to generalizability of the approach, as well as its relevance given that LayerNorm (which performs comparably to the proposed method) is often preferred over BatchNorm+Dropout outweigh the positives. The recommendation is thus to Reject.